# Phosphine Functionalized Cp$^C$ Ligands and Their Metal Complexes

Florian Nährig, Yu Sun and Werner R. Thiel * 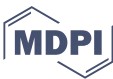

Fachbereich Chemie, RPTU Kaiserslautern-Landau, Erwin-Schrödinger-Straße 54, 67663 Kaiserslautern, Germany
* Correspondence: thiel@chemie.uni-kl.de; Tel.: +49-631-2052752

**Abstract:** Simple nucleophilic aliphatic substitution gives access to mono- and diphosphine ligands with a Cp$^C$ group in the backbone. The monophosphine ligand coordinates to gold(I) via the phosphine site, to thallium(I) via the cyclopentadienyl site and to ruthenium(II) via a combination of both, resulting in an *ansa*-type structure. Coordination with the cyclopentadiene site is not possible for the diphosphine ligand. In this case, monodentate coordination to gold(I) and bidentate coordination to the [PdCl(μ$^2$-Cl)]$_2$, the [Rh(CO)(μ$^2$-Cl)]$_2$, and the Rh(CO)Cl fragment is observed, showing the variability in coordination modes possible for the long-chain diphosphine ligand. Ligands and complexes were characterized by means of NMR and IR spectroscopy, elemental analysis and X-ray structure analysis.

**Keywords:** cyclopentadienyl ligand; phosphine ligand; rhodium; palladium; gold; thallium

## 1. Introduction

Both cyclopentadienyl derivatives and phosphines are among the most frequently used ligands in organometallic chemistry. It is therefore not surprising that Charrier and Mathey published the first example of a phosphine-functionalized cyclopentadiene as early as 1978. They obtained it by reacting sodium cyclopentadienide and chloromethyldiphenylphosphine [1]. The combination of a cyclopentadiene-derived nucleophile with an electrophile bearing a phosphine site remains the most important route to phosphine-functionalized cyclopentadienes. Alternatively, these compounds are accessible by nucleophilic addition of a phosphide to a fulvene or to a spiro [2.4]hepta-4,6-diene derivative [2–5]. Based on these strategies, a large number of such compounds have become accessible in recent decades.

When both the cyclopentadienyl and the phosphine site coordinate to one transition metal center, the resulting compounds exhibit increased stability compared to classical complexes with unfunctionalized cyclopentadienyl ligands, because the η$^5$→η$^3$→η$^1$ shift of the cyclopentadienyl ligand is suppressed by the *ansa*-bridging of the two donors. In addition, the phosphine donor remains in the immediate vicinity of the metal center after dissociation, which has a further stabilizing effect.

Among all phosphine functionalized cyclopentadienyl derivatives, ferrocenyl phosphines have probably attracted the most attention, in particular due to their use as mono- or bidentate ligands in transition metal catalyzed reactions [6–16]

Recently, we presented the $C_2$-symmetric and thus chiral cyclopentadiene derivative (Cp$^C$)$^{-1}$ (**1**$^{-1}$) as a ligand for early and late transition metal complexes [17–19]. The corresponding cyclopentadiene Cp$^C$H (**1**) is accessible in a few steps with high yield from dibenzosuberenone. In parallel to the transition metal chemistry of compound **1**, we searched for new derivatives functionalized at its cyclopentadiene core. By reacting the anion (Cp$^C$)$^{-1}$ (**1**$^{-1}$) with oxygen, the corresponding ketone is accessible in good yield. With this η$^4$-coordinating dienone, we were able to synthesize ruthenium and iron compounds whose structures and reactivities correspond to those of the Shvo and Knölker catalysts, respectively [20]. Here, we now report on novel phosphine-functionalized derivatives of **1** and their use as ligands in transition metal chemistry.

## 2. Materials and Methods

Chemicals were purchased from the following suppliers: ABCR GmbH, Fisher Scientific GmbH, Merck KGaA and Strem Chemicals GmbH. All commercially available starting materials were used without further purification. Compounds sensitive to air or moisture were handled and reacted under exclusion of oxygen and water in suitable Schlenk tubes. The solvents dichloromethane, diethyl ether, *n*-pentane and toluene were dried with a MBraun MB-SPS solvent drying system and degassed by passing nitrogen for 10 min. Acetonitrile and tetrahydrofuran were dried according to standard methods [21]. Glassware was heated three times with a heat gun under vacuum and refilled with dry nitrogen before use. When necessary, the purification of compounds was carried out by column chromatography on the CombiFlash Rf200 instrument from Teledyne Isco with pre-packed RediSept® columns. NMR spectra were recorded using Avance 400 and 600 devices from Bruker Corporation at a temperature of 293 K (20 °C). The resonances were only partly assignable (numbering according to Scheme 1). Air- or moisture-sensitive compounds were measured under an atmosphere of nitrogen using an NMR tube with a Teflon cap from VWR International GmbH. The anhydrous deuterated solvents $CD_3CN$, $C_6D_6$, $CDCl_3$ and $CD_2Cl_2$ were dried according to standard methods, re-condensed and stored under an atmosphere of nitrogen in Schlenk tubes. The evaluation of the NMR spectra was carried out with the software MestReNova 6.0.2-5475 © of Mestrelab Research. The infrared spectra were recorded on a Perkin-Elmer FT-ATR-IR 100 spectrometer with an ATR cell with diamond-coated zinc selenide windows. The IR spectra were processed using the Perkin-Elmer Spectrum 6.3.5 software from and the OriginLab Corporation software OriginPro 8G. Elemental analyses were measured out in the Analytical Laboratory of the Department of Chemistry at the TU Kaiserslautern. Compounds sensitive to air or moisture were filled in a glove box into tin capsules and sealed under an atmosphere of argon. The measurement of the elemental analyses was carried with a Vario Micro Cube analyser from Elementar-Analysetechnik. $Cp^CH$ (**1**) was synthesized according to a published procedure [17]. (tht)AuCl, TlOEt, $[(\eta^6\text{-}C_6H_6)RuCl_2]_2$, $(CH_3CN)_2PdCl_2$ and $[(CO)_2Rh(\mu^2\text{-}Cl)]_2$ were obtained commercially.

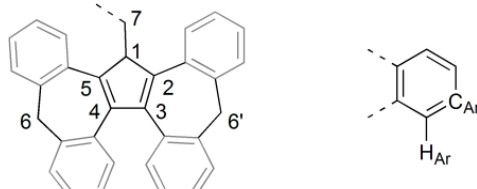

**Scheme 1.** Numbering for the assignment of NMR resonances (see below) of the $Cp^C$ complexes.

$Cp^CHBzPPh_2$ (**2**): A total of 2.17 mL (3.47 mmol) of *n*-butyllithium (1.6 M solution in *n*-hexane) was slowly added at 0 °C to a solution of 1.24 g of $Cp^CH$ (**1**) [17] (3.14 mmol) in 10 mL of toluene. The reaction mixture was stirred for 2 h at this temperature and then was slowly warmed to room temperature. During this time, the precipitation of a colorless solid was observed, which was dissolved by the addition of 1 mL of tetrahydrofurane. Then, 976 mg (3.14 mmol) of [2-(chloromethyl)phenyl](diphenyl)phosphane [22,23], dissolved in 10 mL of toluene, was slowly added at 0 °C. The reaction mixture was stirred for additional 2 h at this temperature. During this time, a color change from orange to dark red was observed and the reaction mixture was stirred for another 18 h at room temperature. A total of 15 mL of water was added, and the organic phase was separated, washed three times with 5 mL of water and dried over $MgSO_4$. The solvent was removed, and the crude red–brown product was dried under vacuum. Purification was carried out by column chromatography (MPLC) with a mixture of *n*-hexane and ethyl acetate (99:1). Compound **2** was obtained slightly contaminated as a light pink solid in the first fraction. As a minor product (16%), compound **3** can be isolated from the second fraction. Compound **2** was finally purified by precipitation from a saturated dichloromethane solution with

*n*-pentane. Yield: 1.23 g (59%) of a colorless solid. Elemental analysis calcd. for $C_{50}H_{37}P$ (668.82 g/mol): C 89.79, H 5.58; found: C 89.87, H 5.70%. $^1$H NMR (400 MHz, CDCl$_3$): δ 7.55 (d, $^3J_{HH}$ = 7.9 Hz, 1H, H$_{Ar}$), 7.34–7.28 (m, 2H, H$_{Ar}$), 7.25–7.05 (m, 15H, H$_{Ar}$), 6.98 (dt, $^3J_{HH}$ = 15.5, 7.2 Hz, 5H, H$_{Ar}$), 6.89–6.84 (m, 1H, H$_{Ar}$), 6.84–6.79 (m, 2H, H$_{Ar}$), 6.76 (t, $^3J_{HH}$ = 6.9 Hz, 2H, H$_{Ar}$), 6.60 (d, $^3J_{HH}$ = 7.1 Hz, 2H, H$_{Ar}$), 4.69 (t, $^3J_{HH}$ = 5.1 Hz, 1H, H1), 4.07 (dt, $^2J_{HH}$ = 16.7, $^3J_{HH}$ = 3.3 Hz, $^4J_{PH}$ = 3.3 Hz, 1H, H7), 3.80 (d, $^2J_{HH}$ = 12.7 Hz, 1H, H6 or H6′), 3.75–3.63 (m, 3H, H6 or H6′ and H7), 3.33 (d, $^3J_{HH}$ = 12.6 Hz, 1H, H6 or H6′). $^{13}$C NMR (101 MHz, CDCl$_3$): δ 147.9, 144.9, 143.9, 143.1, 142.9, 141.6, 141.0, 139.8, 139.6, 136.8 (d, $J_{PC}$ = 7.4 Hz), 136.7 (d, $J_{PC}$ = 7.8 Hz), 136.2, 136.0, 134.2, 134.2, 134.1, 134.0, 133.9, 133.1, 132.9, 132.66, 132.1, 129.2, 128.9, 128.9, 128.8, 128.7, 128.6, 128.0, 128.0, 127.9, 127.4, 127.0, 126.6, 126.5, 126.3, 125.1, 124.8, 55.1 (C1), 42.2 (C6 or C6′), 41.8 (C6 or C6′), 34.6 (d, $^3J_{PC}$ = 24.1 Hz, C7). $^{31}$P NMR (162 MHz, CDCl$_3$): δ −15.44 (s). IR (ATR, cm$^{-1}$): ṽ 3059 m, 2962 m, 2921 m, 2872 w, 1584 m, 1485 m, 1464 m, 1432 s, 1333 m, 1304 m, 1277 m, 1213 m, 1193 m, 1144 m, 1124 m, 1090 m, 1027 m, 940 m, 779 m, 765 s, 746 vs, 717 s, 697 vs, 674 s.

Cp$^C$(BzPPh$_2$)$_2$ (**3**): A total of 0.74 mL (1.18 mmol) of *n*-butyllithium (1.6 M in *n*-hexane) wasadded at 0 °C to a solution of 716 mg (1.07 mmol) of **2** in (10 mL) of toluene. The reaction mixture was stirred for 2 h at this temperature and afterwards slowly warmed to room temperature. During this time, the precipitation of a colorless solid was observed, which was dissolved by the addition of 1 mL of tetrahydrofurane. Then, 333 mg (1.07 mmol) of (diphenyl)phosphane [22,23] dissolved in 5 mL of toluene wasadded slowly at 0 °C and the reaction mixture was stirred for another 2 h at this temperature, whereby its color changed from pink to dark red. After warming to room temperature, the mixture was stirred for further 18 h. A total of 5 mL of water was added, the organic phase was separated, washed three times with 5 mL of water, and then dried over MgSO$_4$. The solvent was removed, and the light pink crude product was dried under vacuum. Finally, the crude product was purified by precipitation from a saturated dichloromethane solution with *n*-pentane. Colorless prismatic single crystals were obtained by slow diffusion of *n*-pentane into a saturated toluene solution. Yield: 840 mg (83%) of a colorless solid. Elemental analysis calcd. For $C_{69}H_{52}P_2$ (943.12 g/mol): C 87.87, H 5.56; found: C 87.86, H 5.9%. $^1$H NMR (400 MHz, CDCl$_3$): δ 7.81 (d, $^3J_{HH}$ = 7.9 Hz, 2H, H$_{Ar}$), 7.42–7.30 (m, 18H, H$_{Ar}$), 7.25 (dd, $^3J_{HH}$ = 10.3, 5.5 Hz, 4H, H$_{Ar}$), 7.17 (t, 3 $^3J_{HH}$ = 7.4 Hz, 2H, H$_{Ar}$), 7.09 (dd, $^3J_{HH}$ = 10.6, 4.4 Hz, 4H, H$_{Ar}$), 7.02–6.94 (m, 6H, H$_{Ar}$), 6.90 (t, $^3J_{HH}$ = 7.6 Hz, 2H, H$_{Ar}$), 6.85 (t, $^3J_{HH}$ = 7.6 Hz, 2H, H$_{Ar}$), 6.71 (dd, $^3J_{HH}$ = 7.3, 3.8 Hz, 2H, H$_{Ar}$), 6.50 (d, $^3J_{HH}$ = 7.7 Hz, 2H, H$_{Ar}$), 4.33 (dd, $^2J_{HH}$ = 16.1, $^4J_{PH}$ = 8.0 Hz, 2H, H7), 3.84 (d, $^2J_{HH}$ = 16.1 Hz, 2H, H7), 3.39 (d, $^2J_{HH}$ = 12.5 Hz, 2H, H6), 2.48 (d, $^2J_{HH}$ = 12.4 Hz, 2H, H6). $^{13}$C NMR (101 MHz, CDCl$_3$): δ 147.6, 145.0, 142.4, 141.7, 141.5, 141.1, 137.8, 137.7, 137.3, 137.2, 137.0, 134.4 (d, $J_{PC}$ = 20.2 Hz), 133.9 (d, $J_{PC}$ = 20.1 Hz), 133.3, 133.0, 132.3, 128.9, 128.7, 128.5 (d, $J_{PC}$ = 19.4 Hz), 128.3, 128.1, 127.8, 127.5 (d, $J_{PC}$ = 2.1 Hz), 126.8, 126.6, 126.5, 126.1, 124.7, 65.6 (t, $^4J_{PC}$ = 2.8 Hz, C1), 40.9 (br, C6), 40.1 (dt, $^3J_{PC}$ = 10.7, 5.7 Hz, C7). $^{31}$P NMR (162 MHz, CDCl$_3$): δ −16.22 (s). IR (ATR, cm$^{-1}$): ṽ = 3053 m, 3023 w, 2999 w, 2927 w, 2905 w, 2867 w, 2843 w, 1586 m, 1480 m, 1461 m, 1434 s, 1264 w, 1157 w, 1120 w, 1092 m, 1065 m, 1026 m, 947 w, 904 m, 754 s, 738 vs, 694 vs, 680 m, 656 m.

[Cp$^C$HBzPPh$_2$(AuCl)] (**4**): The reaction was carried out in the absence of light. A total of 222 mg (0.33 mmol) of compound **2** and 136 mg (0.42 mmol) of (tetrahydrothiophene)gold(I) chloride was stirred for 18 h at room temperature in 15 mL of dichloromethane. A total of 30 mL of *n*-pentane was added to precipitate a colorless solid, which was filtered off, washed three times with 15 mL of diethyl ether and three times with 15 mL of *n*-pentane, and finally, dried under vacuum. Yield: 298 mg (78%) of a colorless solid. Prismatic single crystals containing one equivalent of toluene were obtained by slow diffusion of *n*-pentane into a saturated toluene solution. Elemental analysis calcd. for $C_{50}H_{37}AuClP$ (901.22 g/mol): C 66.64, H 4.14, found: C 66.53, H 4.10%. $^1$H NMR (400 MHz, CDCl$_3$): δ 7.66–7.57 (m, 3H, H$_{Ar}$), 7.55–7.48 (m, 2 × $^3J_{HH}$ = 8.8 and 7.6 Hz, 1.9 Hz, 3H, H$_{Ar}$), 7.42–7.27 (m, 9H, H$_{Ar}$), 7.25–7.14 (m, 6H, H$_{Ar}$), 7.11–6.92 (m, 6H, H$_{Ar}$), 6.91–6.85 (m, 1H, H$_{Ar}$), 6.75–6.67 (m, 2H, H$_{Ar}$), 4.65 (t, $^3J_{HH}$ = 4.6 Hz, 1H, H1), 4.51 (dd, $^2J_{HH}$ = 18.3 Hz,

4.7 Hz, 1H, H7), 4.05 (dd, $^2J_{HH}$ = 18.7 Hz, 3.5 Hz, 1H, H7), 3.86 (d, $^2J_{HH}$ = 12.7 Hz, 1H, H6 or H6′), 3.79 (dd, $^2J_{HH}$ = 12.6 Hz, 7.2 Hz, 2H, H6 or H6′), 3.60 (d, $^2J_{HH}$ = 12.5 Hz, 1H, H6 or H6′). $^{13}$C NMR (101 MHz, CDCl$_3$): δ 146.1, 144.2, 143.5, 143.3, 143.2, 142,0 141.6, 140.7, 139.7 (d, $J_{PC}$ = 5.1 Hz), 135.2 (d, $J_{PC}$ = 14.0 Hz), 134.2 (d, $J_{PC}$ = 13.8 Hz), 133.4, 132.4 (d, $J_{PC}$ = 2.5 Hz), 132.3, 132.2, 132.1 (d, $J_{PC}$ = 1.9 Hz), 132.1, 131.8, 129.6 (d, $J_{PC}$ = 12.0 Hz), 129.3, 129.2, 129.0, 129.0, 128.9, 128.4, 128.1, 128.0, 128.0, 127.9, 127.9, 127.7, 127.7, 127.5, 127.4, 127.0, 126.9, 126.9, 126.8, 126.7, 126.6 (d, $J_{PC}$ = 9.8 Hz), 126.2, 125.0 (d, $J_{PC}$ = 10.0 Hz), 53.8 (C1), 41.9 (C6 or C6′), 41.8 (C6 or C6′), 32.4 (d, $J_{PC}$ = 15.8 Hz, C7). $^{31}$P NMR (162 MHz, CDCl$_3$): δ 24.4. IR (ATR, cm$^{-1}$): ṽ 3054 w, 3014 w, 2980 m, 2886 w, 2831 w, 1589 w, 1480 m, 1436 s, 1351 w, 1250 w, 1185 w, 1158 w, 1121 w, 1099 m, 1039 w, 998 w, 945 w, 908 w, 771 s, 744 vs, 711 s, 691 vs.

[(η$^5$-Cp$^C$HBzPPh$_2$)Tl] (**5**): The reaction was carried out in the absence of light. A total of 1.33 g (1.99 mmol) of **2** was dissolved in 30 mL o dry benzene. A total of 874 mg (350 mmol) of thallium(I) ethanolate was added and the mixture was stirred for 30 min at room temperature. A bright yellow solid precipitated, which was filtered off, washed three times with 40 mL of benzene and dried under vacuum. Yield 1.44 g (83%) of a bright yellow microcrystalline solid, which was recrystallized by cooling a hot saturated solution in THF to room temperature. Elemental analysis calcd. for C$_{50}$H$_{36}$PTl (872.18 g/mol): C 68.85, H 4.16; found: C 68.82, H 4.26%. $^1$H NMR (600 MHz, DMSO-d$^6$): δ 7.58–7.53 (m, 1H, H$_{Ar}$), 7.46–7.38 (m, 6H, H$_{Ar}$), 7.30 (d, $^3J_{HH}$ = 7.2 Hz, 3H, H$_{Ar}$), 7.26 (t, $^3J_{HH}$ = 7.1 Hz, 3H H$_{Ar}$), 7.17 (t, $^3J_{HH}$ = 7.2 Hz, 2H, H$_{Ar}$), 7.15–7.12 (m, 1H, H$_{Ar}$), 7.10–7.05 (m, 2H, H$_{Ar}$), 6.98–6.83 (m, 9H, H$_{Ar}$), 6.78–6.74 (m, 2H, H$_{Ar}$), 6.61 (dd, 2 × $^3J_{HH}$ = 6.9, 4.8 Hz, 1H, H$_{Ar}$), 5.05 (d, $^2J_{HH}$ = 18.4 Hz, 1H, H7), 4.39 (d, $^2J_{HH}$ = 18.4 Hz, 1H, H7), 4.14 (d, $^3J_{HH}$ = 11.6 Hz, 1H, H6 or H6′), 3.83 (br, 2H, H6 or H6′), 3.74 (d, $^3J_{HH}$ = 10.8 Hz, 1H, H6 or H6′). $^{13}$C NMR (151 MHz, DMSO-d$^6$): δ 148.0, 147.8, 139.8, 139.36, 138.6, 136.4, 136.3, 136.2, 135.6, 134.4, 134.3, 133.6 (d, $J_{PC}$ = 19.7 Hz), 133.4 (d, $J_{PC}$ = 19.5 Hz), 131.8, 130.9, 130.2, 128.9, 128.8, 128.8, 128.8, 128.7, 128.5, 128.2 (d, $J_{PC}$ = 3.9 Hz), 128.0, 127.6, 127.1, 126.9, 126.3, 125.4, 125.1, 124.8, 124.5, 124.4, 124.4, 123.8, 123.8, 123.7, 123.5, 123.0, 122.8, 121.1, 118.7, 115.5 (d, $J_{PC}$ = 3.7 Hz, C1), 43.1 (C6 or C6′), 41.3 (C6 or C6′), 30.1 (d, $J_{PC}$ = 24.3 Hz, C7). $^{31}$P NMR (243 MHz, DMSO-d$^6$): δ −15.6.

[(η$^5$-Cp$^C$BzPPh$_2$)Ru(NCMe)$_2$]PF$_6$ (**6**): A total of 579 mg (0.66 mmol) of **5**, 150 mg (0.30 mmol) of [(η$^6$-C$_6$H$_6$)RuCl$_2$]$_2$ and 140 mg (0.76 mmol) of KPF$_6$ were dissolved in 20 mL of acetonitrile and was stirred for 18 h at room temperature, while the color of the mixture changed from yellow to orange. The precipitated solid (TlCl, KCl) was filtered off and the volume of the solution was reduced to about 5 mL by removing the solvent under vacuum. By the addition of 30 mL of diethyl ether, an orange colored solid precipitated, which was filtered off, washed three times with 15 mL of diethyl ether and three times with 15 mL of n-pentane, and finally, dried under vacuum. Yield: 530 mg (80%) of an orange colored solid. Elemental analysis calcd. for C$_{54}$H$_{42}$F$_6$N$_2$P$_2$Ru (1036.98 g/mol): C 65.12, H 4.25, N 2.81; found: C 64.89, H 4.48, N 2.74. $^1$H NMR (400 MHz, CD$_3$CN): δ 7.63–7.58 (m, 2H, H$_{Ar}$), 7.52–7.49 (m, 3H, H$_{Ar}$), 7.48–7.42 (m, 3H, H$_{Ar}$), 7.42–7.34 (m, 4H, H$_{Ar}$), 7.31 (d, $^3J_{HH}$ = 7.7 Hz, 1H, H$_{Ar}$), 7.27–7.21 (m, 3H, H$_{Ar}$), 7.19–7.13 (m, 3H, H$_{Ar}$), 7.11–6.99 (m, 6H, H$_{Ar}$), 6.96–6.91 (m, 1H, H$_{Ar}$), 6.90–6.82 (m, 2H, H$_{Ar}$), 6.63 (td, $^3J_{HH}$ = 7.7, $^4J_{HH}$ = 0.8 Hz, 1H, H$_{Ar}$), 6.17 (d, $^3J_{HH}$ = 7.7 Hz, 1H, H$_{Ar}$), 4.47 (d, $^2J_{HH}$ = 13.1 Hz, 1H, H6 or H6′), 4.38 (dd, $^2J_{HH}$ = 18.0, 4 J PH = 6.8 Hz, 1H, H7), 4.18 (d, $^2J_{HH}$ = 12.8 Hz, 1H, H6 or H6′), 4.12 (d, $^2J_{HH}$ = 17.9 Hz, 1H, H7), 3.78 (t, $^2J_{HH}$ = 13.2 Hz, 2H, H6 or H6′), 1.96 (s, 6H, CH$_3$CN). $^{13}$C NMR (101 MHz, CD$_3$CN): δ = 147.0, 146.8, 144.5, 143.9, 143.7 (d, $J_{PC}$ = 2.5 Hz), 142.1, 135.1 (d, $J_{PC}$ = 3.2 Hz), 134.9, 134.8 (d, $J_{PC}$ = 3.2 Hz), 134.4, 133.8 (d, $J_{PC}$ = 10.1 Hz), 133.2, 132.9 (d, $J_{PC}$ = 10.7 Hz), 132.6, 132.5 (d, $J_{PC}$ = 17.0 Hz), 132.0 (d, $J_{PC}$ = 1.8 Hz), 131.9 (d, $J_{PC}$ = 1.8 Hz), 131.2, 131.1 (d, $J_{PC}$ = 1.9 Hz), 130.7 (d, $J_{PC}$ = 1.7 Hz), 130.5, 130.3, 130.2, 129.5, 129.4, 129.3, 129.2, 129.1, 128.9, 128.5 (d, $J_{PC}$ = 8.7 Hz), 128.4, 128.3, 128.3, 127.1, 126.5, 126.5, 126.3 (d, $J_{PC}$ = 9.0 Hz), 126.0, 105.6 (d, $J_{PC}$ = 3.5 Hz, C1), 92.4 (d, $J_{PC}$ = 9.6 Hz, C2-C5), 92.0 (C2-C5), 87.8 (C2-C5), 75.0 (C2-C5), 41.9 (C6 or C6′), 41.0 (C6 or C6′), 30.6 (d, $J_{PC}$ = 5.3 Hz, C7). $^{31}$P NMR (162 MHz, CD$_3$CN): δ 51.7 (s, PPh$_2$), −144.6 (hept., $^1J_{PF}$ = 706.3 Hz, PF$_6^-$). $^{19}$F NMR (376 MHz, CD$_3$CN): δ −72.9 (d, $^1J_{PF}$ = 706.1 Hz, PF$_6^-$). ESI-MS (CD$_3$CN): *m/z* found (calcd.)

810.13 (810.19, $[C_{50}H_{36}PRu(CH_3CN)]^+$), 769.09 (769.16, $[C_{50}H_{36}PRu]^+$). IR (ATR, cm$^{-1}$): $\tilde{\nu}$ 3056 w, 3005 w, 2963 w, 1926 w, 2853 w, 2038 vw, 1599 w, 1483 m, 1435 s, 1371 w, 1261 w, 1162 w, 1093 m, 1039 w, 1028 w, 950 w, 919 w, 876 m, 834 vs, 754 s, 747 s, 738 s, 719 m, 696 s, 674 m.

$[Cp^C(BzPPh_2(AuCl)_2)]$ (**7**): The reaction was carried analogous to the synthesis of the gold(I) complex **4** using 239 mg (0.25 mmol) of the diphosphine derivative **3** and 171 mg (tht)AuCl. After stirring the mixture for 18 h at room temperature in 15 mL of $CH_2Cl_2$, the colorless precipitate was filtered of, washed four times with 15 mL of diethyl ether and then dried under vacuum. Prismatic single crystals containing one equivalent of toluene were obtained by slow diffusion of *n*-pentane into a saturated toluene solution. Yield: 313 mg (83%) of a colorless solid. Due to the poor solubility of compound **7** in all organic solvents, it was impossible to obtain a $^{13}C$ NMR spectrum. Elemental analysis calcd. for $C_{69}H_{52}Au_2Cl_2P_2$ (1407.94 g/mol): C 58.86, H 3.72; found: C 58.62, H 3.83%. $^1H$ NMR (400 MHz, CDCl$_3$): $\delta$ 7.61–7.56 (m, $^3J_{HH}$ = 7.0 Hz, 2H, H$_{Ar}$), 7.52–7.45 (m, 8H, H$_{Ar}$), 7.40–7.27 (m, 20H, H$_{Ar}$), 7.22–7.17 (m, $^3J_{HH}$ = 9.6, 3.9 Hz, 4H, H$_{Ar}$), 7.08–7.04 (m, 4H, H$_{Ar}$), 6.91–6.83 (m, 4H, H$_{Ar}$), 6.60 (d, $^3J_{HH}$ = 7.6 Hz, 2H, H$_{Ar}$), 4.01 (d, $^2J_{HH}$ = 18.2 Hz, 2H, H7 or H7′), 3.65 (d, $^2J_{HH}$ = 17.1 Hz, 2H, H7 or H7′), 3.51 (d, $^2J_{HH}$ = 12.6 Hz, 2H, H6 or H6′), 2.89 (d, $^2J_{HH}$ = 12.5 Hz, 2H, H6 or H6′). $^{31}P$ NMR (162 MHz, CDCl$_3$): $\delta$ 24.9. IR (ATR, cm$^{-1}$): $\tilde{\nu}$ 3058 w, 3015 w, 2948 w, 2925 w, 2842 w, 1588 w, 1480 m, 1438 s, 1310 w, 1261 m, 1179 w, 1101 s, 1068 m, 1029 m, 999 m, 903 w, 847 w, 793 m, 757 vs, 743 vs, 689 m.

$[Cp^C(BzPPh_2)_2(PdCl(\mu^2\text{-}Cl))_2)]$ (**8**): 183 mg (0.19 mmol) of ligand **3** and 101 mg (0.39 mmol) of $(CH_3CN)_2Pd(Cl)_2$ were dissolved in 6 mL of $CH_2Cl_2$. The mixture was stirred for 18 h at room temp. Then, 10 mL of *n*-pentane were added. An orange colored solid precipitated, which was filtered off, washed three times with 5 mL of diethyl ether and three times with 5 mL of *n*-pentane and dried under vacuum. Yield: 197 mg (78%) of a temperature-sensitive orange colored solid, which was stored at T < 10 °C. Single crystals were obtained by slow diffusion of *n*-pentane into a saturated solution of the compound in toluene. Elemental analysis calcd. for $C_{69}H_{52}Cl_4P_2Pd_2$ (1297.77 g/mol): C 63.86, H 4.04; found: C 63.56, H 3.93%. $^1H$ NMR (400 MHz, CDCl$_3$): $\delta$ 7.87 (dd, $^3J_{HH}$ = 12.5, 7.6 Hz, 4H, H$_{Ar}$), 7.83–7.73 (m, 4H, 2 × H$_{Ar}$ and 2×H7 or H7′), 7.60 (d, $^3J_{HH}$ = 8.2 Hz, 2H, H$_{Ar}$), 7.55–7.50 (m, 2H, H$_{Ar}$), 7.45 (d, $^3J_{HH}$ = 7.5 Hz, 2H, H$_{Ar}$), 7.42–7.37 (m, 6H, H$_{Ar}$), 7.33–7.29 (m, 2H, H$_{Ar}$), 7.25–7.21 (m, 2H, H$_{Ar}$), 7.16–7.07 (m, 4H, H$_{Ar}$), 7.05–7.00 (m, 2H, H$_{Ar}$), 6.98–6.88 (m, 9H, H$_{Ar}$), 6.85 (dd, $^3J_{HH}$ = 6.1, 0.7 Hz, 1H, H$_{Ar}$), 6.83–6.78 (m, 2H, H$_{Ar}$), 6.62 (dd, $^3J_{HH}$ = 12.5, 7.9 Hz, 4H, H$_{Ar}$), 4.80 (d, $^2J_{HH}$ = 17.7 Hz, 2H, H7 or H7′), 3.89 (d, $^2J_{HH}$ = 12.3 Hz, 2H, H6 or H6′), 3.82 (d, $^2J_{HH}$ = 12.5 Hz, 2H, H6 or H6′). $^{13}C$ NMR (101 MHz, CDCl$_3$): $\delta$ 147.9, 144.3, 141.6 (d, $J_{PC}$ = 10.2 Hz), 141.0, 140.6, 137.8 (d, $J_{PC}$ = 11.6 Hz), 133.0 (d, $J_{PC}$ = 9.6 Hz), 132.3, 131.8, 130.4, 129.1, 128.8, 128.7, 128.6, 128.5, 128.4, 128.3, 128.0, 127.7, 127.4, 127.3, 127.2, 126.9 (d, $J_{PC}$ = 21.3 Hz), 126.7, 126.6, 126.4, 126.3 (d, $J_{PC}$ = 3.5 Hz), 125.1, 63.8 (C1), 48.2 (d, $J_{PC}$ = 11.4 Hz, C7 or C7′), 41.88 (C6 or C6′). $^{31}P$ NMR (162 MHz, CDCl$_3$): $\delta$ 25.4 (s). IR (ATR, cm$^{-1}$): $\tilde{\nu}$ 3056 m, 2945 w, 2892 w, 2836 w, 1589 w, 1480 m, 1461 w, 1435 s, 1327 w, 1188 w, 1161 w, 1092 s, 999 w, 902 w, 850 w, 757 s, 741 s, 706 s, 690 s.

$[Cp^C(BzPPh_2)_2)(Rh(\mu^2\text{-}Cl(CO))_2]$ (**9**): A total of 107 mg (0.11 mmol) of ligand **3** and 45.3 mg (0.11 mmol) of $[(CO)_2Rh(\mu^2\text{-}Cl)]_2$ was dissolved in 6 mL of dichloromethane. The mixture was stirred for 2 h at room temperature. Then, 10 mL of n-pentane was added. A bright yellow solid precipitated, which was filtered off, washed three times with 5 mL of *n*-pentane and dried under vacuum. Yield: 122 mg (85%) of a bright yellow microcrystalline solid, which was stored at temperatures below 10 °C to prevent decomposition. Yellow prismatic single crystals were obtained by slow diffusion of *n*-pentane into a saturated solution of the compound in toluene. Elemental analysis calcd. for $C_{71}H_{52}Cl_2O_2P_2Rh_2$ (1275.84 g/mol): C 66.84, H 4.11; found: C 67.10, H 4.42%. $^1H$ NMR (600 MHz, CDCl$_3$): $\delta$ 8.47 (d, $^3J_{HH}$ = 7.8 Hz, 1H, H$_{Ar}$), 7.80–7.70 (m, 2H, H$_{Ar}$), 7.65 (dd, $^3J_{HH}$ = 7.0, $^3J_{HH}$ = 5.6 Hz, 1H, H$_{Ar}$), 7.55–7.51 (m, $^3J_{HH}$ = 11.1, $^3J_{HH}$ = 4.1 Hz, 1H, H$_{Ar}$), 7.47–7.34 (m, 12H, H$_{Ar}$), 7.32 (t, $^3J_{HH}$ = 7.5 Hz, 1H, H$_{Ar}$), 7.31–7.25 (m, 6H, H$_{Ar}$), 7.24–7.21 (m, 2H, H$_{Ar}$), 7.18 (t, $^3J_{HH}$ = 7.7 Hz, 1H, H$_{Ar}$), 7.12–7.04 (m, 5H, 3 × H$_{Ar}$ and 2 × H7 or H7′), 7.02–6.95 (m, 5H, H$_{Ar}$), 6.90 (d, $^3J_{HH}$ = 7.6 Hz, 1H, H$_{Ar}$), 6.89–6.82 (m,

$^3J_{HH}$ = 8.3 Hz, 2H, H$_{Ar}$), 6.81 (d, $^3J_{HH}$ = 7.9 Hz, 1H, H$_{Ar}$), 6.71 (dd, $^3J_{HH}$ = 11.8, $^3J_{HH}$ = 7.7 Hz, 1H, H$_{Ar}$), 6.69–6.61 (m, 4H, H$_{Ar}$), 4.03 (d, $^2J_{HH}$ = 12.7 Hz, 1H, H6 or H6′), 3.87 (d, $^2J_{HH}$ = 12.8 Hz, 1H, H6 or H6′), 3.77–3.69 (m, 2H, H6 or H6′ and H7 or H7′), 3.64 (d, $^2J_{HH}$ = 12.7 Hz, 1H, H6 or H6′), 3.45 (d, $^2J_{HH}$ = 18.9 Hz, 1H, H7 + H7′). $^{13}$C NMR (151 MHz, CDCl$_3$): δ 183.2 (dd, $^1J_{RhC}$ = 78.0, $^2J_{PC}$ = 20.6 Hz, CO), 182.3 (dd, $^1J_{RhC}$ = 78.8, $^2J_{PC}$ = 21.1 Hz, CO), 148.0 (d, $J_{PC}$ = 38.4 Hz), 144.6 (d, $J_{PC}$ = 32.7 Hz), 142.7 (d, $J_{PC}$ = 10.0 Hz), 142.2 (d, $J_{PC}$ = 9.6 Hz), 141.9, 141.1, 141.0, 140.7, 136.5 (b), 135.6 (d, $J_{PC}$ = 11.6 Hz), 133.8 (d, $J_{PC}$ = 10.3 Hz), 132.8, 132.6, 132.5, 132.5, 132.2, 132.1, 132.0, 132.0, 131.8, 131.5, 131.0 (d, $J_{PC}$ = 1.5 Hz), 131.0, 130.9, 130.8 (d, $J_{PC}$ = 1.8 Hz), 130.2, 130.0, 129.9, 129.4, 129.2, 129.1, 128.9, 128.88, 128.8, 128.8, 128.6, 128.5, 128.3, 128.2, 128.1, 128.0, 127.8 (d, $J_{PC}$ = 4.3 Hz), 127.7 (d, $J_{PC}$ = 4.2 Hz), 127.5, 127.5, 127.4, 126.9, 126.7, 126.7, 126.1, 126.0, 126.0, 125.7, 125.6, 125.3, 124.9, 63.5 (C1), 46.5 (d, $J_{PC}$ = 15.5 Hz, C7 or C7′), 44.9 (d, $J_{PC}$ = 13.8 Hz, C7 or C7′), 42.1 (C6 or C6′), 41.7 (C6 or C6′). $^{31}$P NMR (162 MHz, CDCl$_3$): δ 39.8 (d, $^1J_{RhP}$ = 171.5 Hz), 35.9 (d, $^1J_{RhP}$ = 175.0 Hz). IR (ATR, cm$^{-1}$): ṽ 3055 w, 3017 w, 2948 w, 2895 w, 2863 w, 2839 w, 1991 vs, 1975 vs, 1590 w, 1482 m, 1462 w, 1435 s, 1327 w, 1310 w, 1187 m, 1160 m, 1119 m, 1093 s, 999 m, 902 m, 855 w, 764 m, 753 s, 737 vs, 720 m, 690 vs, 677 s, 662 m.

[Cp$^C$(BzPPh$_2$)$_2$(RhCl(CO))] (**10**): A total of 217 mg (0.23 mmol) of ligand **3** and 46.1 mg (0.12 mmol) of [(CO)$_2$Rh(μ$^2$-Cl)]$_2$ were dissolved in 10 mL of dichloromethane. The mixture was stirred for 18 h at 45 °C. Then, 10 mL of *n*-pentane was added. A shiny yellow solid precipitated, which was filtered off, was washed three times with 5 mL of diethyl ether and three times with 5 mL of *n*-pentane and was dried under vacuum. Yield: 157 mg (62%) of a shiny yellow solid. Yellow prismatic single crystals were obtained by slow diffusion of *n*-pentane into a saturated solution of the compound in toluene. Elemental analysis calcd. for C$_{70}$H$_{52}$ClOP$_2$Rh (1109.49 g/mol): C 75.78, H 4.72; found: C 75.48, H 4.68. $^1$H NMR (400 MHz, CDCl$_3$): δ 8.11–8.05 (m, 2H, H$_{Ar}$), 7.87 (s, 1H, H$_{Ar}$), 7.58–7.52 (m, 2H, H$_{Ar}$), 7.48–7.44 (m, 4H, H$_{Ar}$), 7.41–7.28 (m, 9H, H$_{Ar}$), 7.24–7.18 (m, 3H, H$_{Ar}$), 7.17–7.05 (m, 5H, H$_{Ar}$), 7.04–6.86 (m, 7H, H$_{Ar}$), 6.85–6.71 (m, 5H, H$_{Ar}$), 6.65–6.59 (m, 2H, H$_{Ar}$), 6.57–6.48 (m, 4H, H$_{Ar}$), 6.23 (d, $^2J_{HH}$ = 16.8 Hz, 1H, H7 or H7′), 5.56 (d, $^2J_{HH}$ = 16.7 Hz, 1H, H7 or H7′), 4.94 (d, $^3J_{HH}$ = 13.5 Hz, 1H, H7 or H7′), 4.09 (q, $^3J_{HH}$ = 12.3 Hz, 2H, H6 or H6′), 3.55 (d, $^2J_{HH}$ = 12.4 Hz, 1H, H7 or H7′), 3.37 (d, $^2J_{HH}$ = 12.4 Hz, 2H, 1 × H6 or H6′ and 1 × H7 or H7′). $^{13}$C NMR (101 MHz, CDCl$_3$): δ 190.0 (d, $^1J_{RhC}$ = 78.5 Hz), 149.3, 147.7, 143.7 (d, $^{xx}J_{PC}$ = 15.8 Hz), 143.6 (d, $J_{PC}$ = 3.5 Hz), 142.3, 141.2, 140.8, 139.8, 138.0, 137.9, 137.6, 137.2, 136.8, 136.0 (d, $J_{PC}$ = 11.9 Hz), 135.4, 135.1, 135.0 (d, $J_{PC}$ = 1.8 Hz), 134.5, 133.9 (d, $J_{PC}$ = 3.7 Hz), 133.7 (d, $J_{PC}$ = 2.6 Hz), 133.3, 133.2, 132.9, 132.8 (d, $J_{PC}$ = 2.0 Hz), 132.4 (d, $J_{PC}$ = 1.6 Hz), 132.2, 132.1, 132.0, 131.9, 131.4 (d, $J_{PC}$ = 3.8 Hz), 131.0, 130.9, 130.5, 130.2 (d, $J_{PC}$ = 4.0 Hz), 129.8 (d, $J_{PC}$ = 1.9 Hz), 129.2, 128.7, 128.6, 128.5 (d, $J_{PC}$ = 2.5 Hz), 128.4, 128.1, 128.0, 128.0, 127.8, 127.6, 127.6, 127.2, 127.1, 126.9 (d, $J_{PC}$ = 2.2 Hz), 126.5, 126.3, 126.2, 126.1, 125.7, 125.0, 124.8, 66.9 (C1), 49.3 (C7 or C7′), 46.8 (d, $J_{PC}$ = 8.7 Hz, C7 or C7′), 42.9 (C6 or C6′), 41.8 (C6 or C6′). $^{31}$P NMR (162 MHz, CDCl$_3$): δ 35.6 (dd, $^2J_{PP}$ = 340.2, $^1J_{RhP}$ = 127.5 Hz), 26.2 (dd, $^2J_{PP}$ = 339.7, $^1J_{RhP}$ = 123.2 Hz). ESI-MS (CD$_2$Cl$_2$): *m/z* found (calcd.) 1073.39 (1073.25, [C$_{70}$H$_{52}$OP$_2$Rh]$^+$). IR (ATR, cm$^{-1}$): ṽ 3056 w, 3010 w, 2947 w, 2837 w, 1961 vs, 1587 w, 1573 w, 1480 m, 1466 m, 1449 m, 1434 s, 1339 w, 1312 w, 1279 w, 1187 m, 1157 w, 1120 w, 1090 s, 1044 w, 1028 w, 999 m, 947 w, 908 w, 872 w, 841 w, 781 m, 748 vs, 690 vs, 673 m.

X-rax structure analysis: Crystal data and refinement parameters were collected and are presented in Tables 1 and 2. All structures were solved using direct method of SIR92 [24], completed by subsequent difference Fourier syntheses, and refined by full-matrix least-squares procedures [25]. Semi-empirical absorption correction from equivalents (Multiscan) was applied to ligand **3**, while analytical numeric absorption correction was carried out to all other complexes [26]. All non-hydrogen atoms were refined with anisotropic displacement parameters. All hydrogen atoms were placed in calculated positions and refined using a riding model. In the original solved structures of complexes **5**, **7**, **8** and **9**, severely disordered and/or partially occupied solvents were also found. The quality of the measured samples and the corresponding collected raw data were limited. Even with the help of

lots of restraints, these disorders could not be treated satisfyingly. To obtain a better understanding of the main structures, the SQUEEZE process integrated in PLATON was used. Additionally, detailed information has been posted in the corresponding final CIF files. CCDC 2247305-2247311 contain the supplementary crystallographic data for this paper. These data can be obtained free of charge from The Cambridge Crystallographic Data Centre via www.ccdc.cam.ac.uk/data_request/cif (accessed on 4 April 2023).

**Table 1.** Crystallographic data, data collection and refinement.

| | **3** | **4** | **5** |
|---|---|---|---|
| empirical formula | $C_{83}H_{68}P$ | $C_{57}H_{45}AuClP$ | $C_{50}H_{36}PTl$ |
| formula weight | 1127.31 | 993.31 | 872.13 |
| crystal size [mm] | $0.408 \times 0.307 \times 0.239$ | $0.244 \times 0.106 \times 0.069$ | $0.313 \times 0.093 \times 0.063$ |
| T [K] | 293(2) | 150(2) | 150(2) |
| $\lambda$ [Å] | 1.54184 | 1.54184 | 1.54184 |
| crystal system | triclinic | triclinic | monoclinic |
| space group | $P\bar{1}$ | $P\bar{1}$ | $P2_1/n$ |
| a [Å] | 14.1471(5) | 10.3433(3) | 18.7154(2) |
| b [Å] | 14.5119(4) | 15.2162(6) | 12.2606(1) |
| c [Å] | 17.7514(6) | 15.5558(7) | 20.4489(2) |
| $\alpha$ [°] | 107.920(3) | 106.743(4) | 90 |
| $\beta$ [°] | 109.661(3) | 107.515(3) | 97.058(1) |
| $\gamma$ [°] | 93.028(3) | 96.040(3) | 90 |
| V [Å³] | 3215.3(2) | 2185.84(16) | 4656.69(8) |
| Z | 2 | 2 | 4 |
| $\rho$ calcd. [g·cm⁻³] | 1.164 | 1.509 | 1.244 |
| $\mu$ [mm⁻¹] | 0.950 | 7.519 | 7.198 |
| $\Theta$-range [°] | 3.251–62.736 | 3.608–62.729 | 3.417–62.775 |
| refl. coll. | 24,391 | 15,048 | 35,775 |
| indep. refl. | 10,246 [$R_{int}$ = 0.0229] | 6947 [$R_{int}$ = 0.0481] | 7447 [$R_{int}$ = 0.0257] |
| data/restr./param. | 10246/0/768 | 6947/0/542 | 7447/0/469 |
| final R indices [I > 2σ(I)] [a] | 0.0522, 0.1455 | 0.0367, 0.0946 | 0.0171, 0.0402 |
| R indices (all data) | 0.0573, 0.1524 | 0.0381, 0.0961 | 0.0179, 0.0407 |
| GooF [b] | 1.036 | 1.082 | 1.029 |
| $\Delta\rho$max/min (e·Å⁻³) | 1.687/−0.283 | 1.444/−1.808 | 0.391/−0.266 |

[a] $R1 = \Sigma||F_o| - |F_c||/\Sigma|F_o|$, $\omega R2 = [\Sigma(F_o^2 - F_c^2)^2/\Sigma\omega F_o2]^{1/2}$. [b] $GooF = [\Sigma\omega(F_o^2 - F_c^2)^2/(n - p)]^{1/2}$.

**Table 2.** Crystallographic data, data collection and refinement.

| | **7** | **8** | **9** | **10** |
|---|---|---|---|---|
| empirical formula | $C_{69}H_{52}Au_2Cl_2P_2$ | $C_{69}H_{52}Cl_4P_2Pd_2$ | $C_{71}H_{52}Cl_2O_2P_2Rh_2$ | $C_{70}H_{52}ClOP_2Rh$ |
| formula weight | 1407.88 | 1297.64 | 1275.78 | 1109.41 |
| crystal size [mm] | $0.216 \times 0.095 \times 0.079$ | $0.199 \times 0.101 \times 0.044$ | $0.272 \times 0.151 \times 0.137$ | $0.233 \times 0.121 \times 0.059$ |
| T [K] | 150(2) | 150(2) | 150(2) | 150(2) |
| $\lambda$ [Å] | 1.54184 | 1.54184 | 1.54184 | 1.54184 |
| crystal system | triclinic | monoclinic | triclinic | monoclinic |
| space group | $P\bar{1}$ | $P2/n$ | $P\bar{1}$ | $P2_1/n$ |
| a [Å] | 13.1508(3) | 21.0446(7) | 13.2483(3) | 13.3016(3) |
| b [Å] | 13.5694(4) | 13.2823(4) | 14.1374(3) | 21.0013(4) |
| c [Å] | 22.2637(5) | 23.4004(9) | 20.4213(4) | 20.0864(4) |
| $\alpha$ [°] | 76.934(2) | 90 | 89.220(2) | 90 |
| $\beta$ [°] | 87.525(2) | 107.894(4) | 73.556(2) | 109.243(2) |

**Table 2.** *Cont.*

| | 7 | 8 | 9 | 10 |
|---|---|---|---|---|
| $\gamma$ [°] | 64.608(3) | 90 | 75.287(2) | 90 |
| V [Å$^3$] | 3489.67(17) | 6224.5(4) | 3541.10(14) | 5297.7(2) |
| Z | 2 | 4 | 2 | 4 |
| $\rho$ calcd. [g·cm$^{-3}$] | 1.340 | 1.385 | 1.197 | 1.391 |
| $\mu$ [mm$^{-1}$] | 9.186 | 7.028 | 5.189 | 3.997 |
| $\Theta$-range [°] | 3.705–62.698 | 3.327–62.748 | 3.238–62.724 | 3.140–62.754 |
| refl. coll. | 27,287 | 24,022 | 49,439 | 21,589 |
| indep. refl. | 11,102 [$R_{int}$ = 0.0308] | 9919 [$R_{int}$ = 0.0550] | 11,257 [$R_{int}$ = 0.0564] | 8461 [$R_{int}$ = 0.0294] |
| data/restr./param. | 11102/0/676 | 9919/0/695 | 11257/0/713 | 8461/0/676 |
| final R indices [I > 2$\sigma$(I)] [a] | 0.0296, 0.0721 | 0.0510, 0.1388 | 0.0310, 0.0790 | 0.0321, 0.0852 |
| R indices (all data) | 0.0349, 0.0739 | 0.0623, 0.1466 | 0.0335, 0.0807 | 0.0362, 0.0886 |
| GooF [b] | 1.037 | 1.082 | 1.023 | 1.032 |
| $\Delta\rho$max/min (e·Å$^{-3}$) | 1.310/−1.262 | 1.253/−0.718 | 0.734/−0.840 | 0.713/−0.518 |

[a] $R1 = \Sigma ||F_o| - |F_c|| / \Sigma |F_o|$, $\omega R2 = [\Sigma(F_o^2 - F_c^2)^2 / \Sigma \omega F_o 2]^{1/2}$. [b] $GooF = [\Sigma\omega(F_o^2 - F_c^2)^2/(n - p)]^{1/2}$.

## 3. Results

### 3.1. Ligand Synthesis and Characterization

Phosphine functionalized derivatives of Cp$^C$H (**1**) were obtained by the deprotonation of the compound with *n*-butyl lithium and treatment of the intermediately generated LiCp$^C$ with 2-[(chloromethyl)phenyl](diphenyl)phosphine (Scheme 2) [22,23] The process follows a protocol published by Li et al. in 2019 for the synthesis of a phosphine-functionalized tetraethylcyclopentadiene [27]. Astonishingly, there are almost no other reports on cyclopentadienyl compounds with a 2-diphenylphosphanylphenylmethyl unit. Knochel et al. reported the synthesis of chiral 1-($S_{Fc}$)-diphenylphosphanyl-2-(o-diphenylphosphanylphenylmethyl)ferrocene via a multi-step process starting with the Friedel–Crafts acylation of ferrocene with 2-bromobenzoyl chloride [28].

**Scheme 2.** Synthesis of ligands **2** and **3**.

Reacting the substrates in a 1:1 ratio with a slight excess of *n*-butyl lithium allowed us to isolate the mono-substituted derivative Cp$^C$HBzPPh$_2$ (**2**) in 59% yield after purification by column chromatography. As a minor product, the bi-functionalized compound

Cp$^C$(BzPPh$_2$)$_2$ (**3**) was obtained in 16% yield. This speaks for a protonation/deprotonation equilibrium between Cp$^C$H (**1**), LiCp$^C$, Cp$^C$HBzPPh$_2$ (**2**) and LiCp$^C$BzPPh$_2$ and a rather high reactivity of LiCp$^C$BzPPh$_2$ towards the electrophile. Accordingly, the treatment of phosphine **2** with another equivalent of *n*-butyl lithium and the electrophile gave the diphosphine **3** in 83% yield.

While the [31]P NMR spectra of **2** and **3** show rather similar chemical shifts of expected values [27,29], their [1]H and [13]C NMR spectra differ largely according to the symmetry of the compounds. In the [1]H NMR spectrum of $C_2$-symmetric **3**, there are four doublets in the aliphatic region, which are assigned to the diastereotopic protons of the two chemically inequivalent methylene groups of the molecule. In contrast, the [1]H and [13]C NMR spectra of $C_1$-symmetric **2** are more complicated. In the aliphatic region of the [1]H NMR spectrum, for example, there are four resonances for the two, now magnetically inequivalent methylene groups bridging the phenyl groups in the "wings" of the Cp backbone. Three further resonances (ABB' spin system) are assigned to the proton at the Cp ring and the protons of the methylene group linking the Cp ring and the triphenyl phosphine moiety. These methylene protons can easily be identified by their coupling pattern, including a small long-range coupling to the phosphorous atom ($^4J_{PH}$ = 3.3 Hz) [29].

In the literature, there are only a few reports of 1,1-diphosphine-functionalized cyclopentadiene derivatives. In 1965, Keough and Grayson succeeded in adding fluorene to vinylphosphonium salts, resulting in dicationic phosphonium species [30]. Inagaki et al. published fluorene derivatives with two phosphinyl moieties directly bound to the sp$^3$ carbon atom of the fluorene backbone [31]. In addition, there are two patents dealing with such compounds [32,33].

By slow diffusion of *n*-pentane into a saturated toluene solution of compound **3**, single crystals suitable for an X-ray structure analysis could be obtained. Compound **3** crystallizes as colorless prisms of the triclinic space group P$\overline{1}$ with two equivalents of toluene in the unit cell. The two phosphine units are found arranged between the wings of the ligand, in an almost orthogonal position to the central cyclopentadiene ring. Figure 1 shows the structure of **3** and summarizes relevant structural parameters.

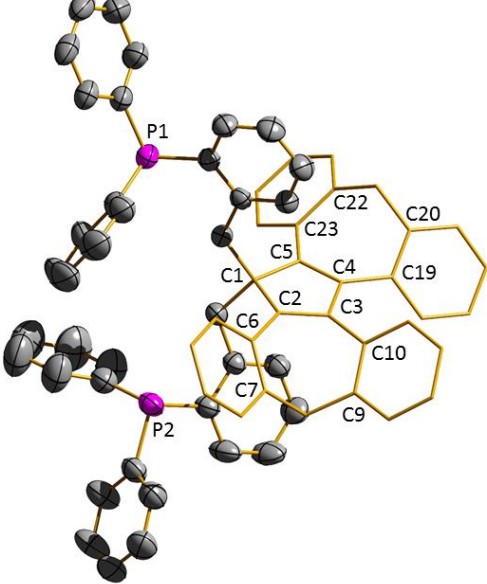

**Figure 1.** Molecular structure of compound **3** in the solid state. The two co-crystallizing molecules of toluene and all hydrogen atoms are omitted for clarity. The central Cp$^C$ system is represented in stick-style for clarity, ellipsoids are drawn at the 50% level. Characteristic bond lengths [Å], angles [°] and dihedral angles [°]: C1-C2 1.525(2), C1-C5 1.532(2), C2-C3 1.358(2), C3-C4 1.476(2), C4-C5 1.359(3), P1···C1···P2 99.25(3), C3-C2-C6-C7 40.5(3), C2-C3-C10-C9 -48.2(3), C4-C5-C23-C22 36.8(3), C5-C4-C19-C20 -45.7(3).

### 3.2. Transition Metal Complexes

While ligand **3** is expected to act like a typical diphosphine, the coordination chemistry of mono-phosphine **2** allows more variations: it may either act as a typical monodentate phosphine, as an $\eta^5$-coordinating Cp-type ligand or may exhibit a combination of these two modes of metal–ligand interactions. To approve the first type of coordination, ligand **2** was treated with [(tht)AuCl] (tht = tetrahydrothiophene, Scheme 3). Gold(I) prefers linear coordination, which avoids involving the Cp ring of **2** in its coordination environment. The according gold(I) complex **4** was obtained in 78% yield as a colorless solid, which is stable under ambient conditions.

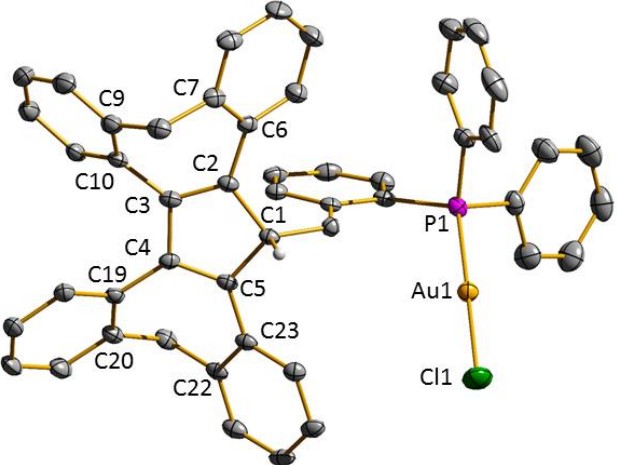

**Scheme 3.** Synthesis of the mononuclear gold(I) complex **4**.

While for typical $^{31}$P NMR resonances of triarylposphine gold(I) chloride, complexes chemical shift of around 32 ppm are reported [34,35], the $^{31}$P NMR resonance of compound **4** appears at 24.4 ppm. We assign this to some shielding effects of the aromatic environment of the Cp$^C$ backbone. The coordination of the gold(I) site in addition leads to a better separation of the aliphatic $^1$H NMR resonances compared to the metal-free ligand **2** (see the Supplementary Materials). Here, a triplet at 4.65 ppm ($^3J_{HH}$ = 4.6 Hz) is assigned to the proton located at the five-membered Cp ring.

By the slow diffusion of *n*-pentane into a saturated toluene solution of compound **4**, single crystals suitable for an X-ray structure analysis could be obtained. Compound **4** crystallizes as colorless prisms of the triclinic space group P$\bar{1}$ with one equivalent of toluene in the unit cell. Figure 2 shows the structure of **4** and summarizes relevant structural parameters.

**Figure 2.** Molecular structure of compound **4** in the solid state. The co-crystallizing molecule of toluene and all hydrogen atoms except the hydrogen atom bound to C1 are omitted for clarity. Ellipsoids are drawn at the 50% level. Characteristic bond lengths [Å], angles [°] and dihedral angles [°]: Au1-P1 2.2427(10), Au1-Cl1 2.2865(10), C1-C2 1.516(5), C1-C5 1.515(5), C2-C3 1.356(6), C3-C4 1.486(5), C4-C5 1.369(6), P1-Au1-Cl1 179.42(4), C3-C2-C6-C7 42.61(7), C4-C5-C23-C22 36.71(7), C2-C3-C10-C9 −45.98(7), C5-C4-C19-C20 −42.35(7).

The crystal structure complex **4** shows the typical linear coordination geometry of gold(I) with an the P1-Au1-Cl1 angle of 179.42(4)°. The measured P1-Au1 (2.2427(10) Å) and Au1-Cl1 (2.2865(10) Å) distances are comparable to the data of (triphenylphosphine)gold(I) chloride and related gold(I) complexes [34,36]. In contrast to many other gold(I) complexes [37–39], there is no hint for a $d^{10}$–$d^{10}$ interaction between the gold(I) sites of neighboring molecules. The phosphine substituent occupies an almost orthogonal position to the central five-membered ring of the $Cp^C$ backbone, minimizing steric repulsions.

We recently published the synthesis of a series of late transition metal complexes wherein $TlCp^C$ is used as the transferring reagent for the $[Cp^C]^-$ ligand [19]. To achieve similar chemistry with a phosphine-functionalized $Cp^C$ derivative, compound **2** was deprotonated in dry benzene with stoichiometric amounts of thallium(I) ethanolate in the absence of light (Scheme 4).

**Scheme 4.** Synthesis of the $Cp^C$ thallium(I) complex **5**.

The thallium(I) compound **5** precipitated immediately after the two reactants were combined and was isolated by filtration with a yield of 83% as a pale yellow solid that can be handled in the air. The $^1$H NMR spectrum of **5** differs clearly from the protonated ligand **2**. In addition to two doublets for the protons of the methylene group linking the Cp ring and the phosphine site ($^2J_{HH}$ = 18.4 Hz), there are four partially superimposed and slightly broadened resonances that are assigned to the four diastereotopic protons methylene groups in the "$Cp^C$-wings". This speaks for a $C_1$ symmetric structure on the NMR time scale and thus for a localization of the thalium(I) cation on one side of the Cp ring. The singlet at −15.56 ppm in the $^{31}$P NMR spectrum is only very weakly shifted towards the lower field compared to the signal of the metal-free ligand **2**.

Crystallization from a saturated solution in the provided yellow, needle-like crystals with the monoclinic space group $P2_1/n$. Figure 3 shows the structure of **5** and summarizes relevant structural parameters.

There are numerous examples in the literature describing the coordination of thallium(I) to aromatic compounds [40–46], while only a few structurally characterized systems with thallium(I)–phosphorus coordination have been published. In 1974, Nakayama et al. published a gas-phase study on a series of thallium(I) complexes and stated that "No thallium(I) complexes with bismuth, arsenic nor phosphorus ligands has been isolated from a liquid phase so far." [47]. This changed during the past two decades, in particular due to the use of weakly coordinating anions [48,49]. The fact that thallium(I) phosphine complexes are hardly accessible as long as there is an alternative for the metal cation is confirmed by the solid-state structure of compound **5**. Compound **5** forms chains with thallium(I) centres bridging two $(Cp^C)^{-1}$ moieties, as shown in Figure 3, and there is no interaction with the phosphine site. In addition to the coordination to the Cp site, thallium(I) also coordinates to two of the six-membered rings, the phenylene ring bound to the phosphorus atom and one of the phenylene rings of the $Cp^C$ moiety. Since the covalency of the bonds between the thallium(I) and the π-donating moieties is low, the Tl-C bond lengths differ largely as documented in the caption of Figure 3 exemplarily for the Cp unit. This allows the adoption of the positioning of the cation to the steric requirements of the solid-state structure. According to the electrostatic interaction between the Cp site and the thallium(I)

cation, the Cp-Tl distance (2.6131(4) Å) is much shorter than the Ar-Tl distances (3.3139(4) and 3.2030(4) Å).

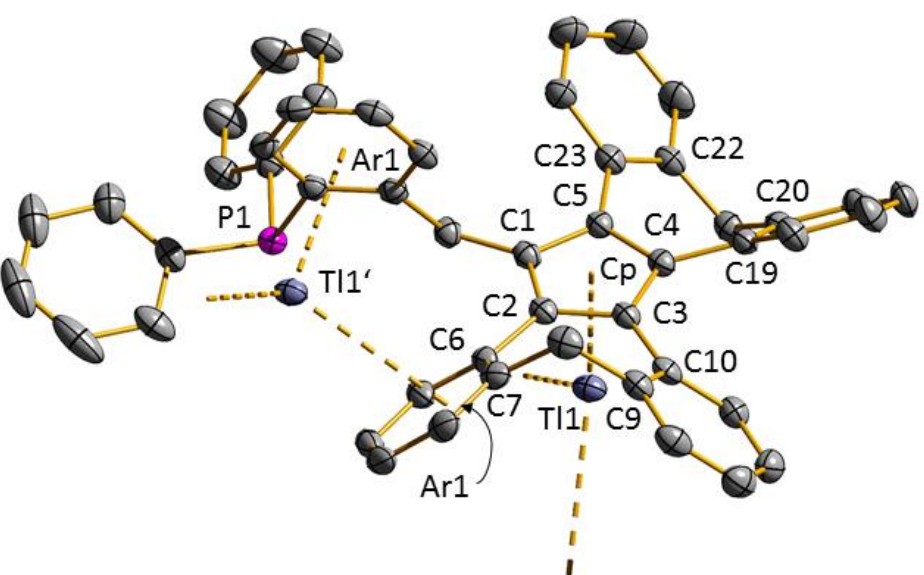

**Figure 3.** Molecular structure of compound **5** in the solid state. All hydrogen atoms are omitted for clarity, ellipsoids are drawn at the 50% level. Characteristic bond lengths [Å], angles [°] and dihedral angles [°]: C1-Tl1 2.8405(18), C2-Tl1 2.8137(18), C3-Tl1 2.8799(17), C4-Tl1 2.9341 (17), C5-Tl1 2.9342 (17), Cp-Tl1 2.6131(3), Ar1-Tl2 3.3139(4), Ar2-Tl2 3.2030(4), C1-C2 1.419(3), C1-C5 1.418(3), C2-C3 1.433(3), C3-C4 1.428(3), C4-C5 1.431(3), Cp-Tl1-Ar1 124.770(9), Cp-Tl1-Ar2 132.793(11), Ar1-Tl-Ar2 95.279(10), C3-C2-C6-C7 41.8(3), C4-C5-C23-C22 44.0(3), C2-C3-C10-C9 −37.3(2), C5-C4-C19-C20 −42.8(3); Cp denotes the centroid of the five-membered cyclopentadienyl ring. Ar1 and Ar2 denote the centroids of the two six-membered rings, which undergo coordination to the thallium(I) cation.

In 2003, Salzer et al. developed a protocol for the preparation of chiral cyclopenta-dienide ruthenium(II) complexes with an *ansa*-bridging phosphine group and two labile bound acetonitrile ligands) [50,51]. They started from cyclopentadiene derivatives bearing a chiral phosphine substituent and $[RuCl(\mu^2\text{-}Cl)(\eta^3\text{:}\eta^3\text{-}C_{10}H_{16})]_2$, used $Li_2CO_3$ as the base and $KPF_6$ to provide a weakly coordinating counter anion. Due to our good experiences with $TlCp^C$ as a $[Cp^C]^{-1}$ transfer reagent [19], we applied a combination of compound **5** and the ruthenium(II) precursor $[(\eta^6\text{-}C_6H_6)RuCl_2]_2$ to generate an analogous $Cp^C$ derivative (Scheme 5).

**Scheme 5.** Synthesis of the $Cp^C$ ruthenium(II) complex **6**.

The reaction was carried out at room temperature in acetonitrile in the presence of a stoichiometric amount of KPF$_6$. After work-up, compound **6** was isolated in 80% yield as an orange-colored solid, which was characterized by means of NMR and IR spectroscopy, mass spectrometry and elemental analysis. Tiny crystals of insufficient quality were obtained by slow diffusion of diethylether into a solution of compound 6 in acetonitrile. The presence of the two acetonitrile ligands was confirmed by a singlet resonance at 1.96 ppm in the $^1$H NMR spectrum and by an absorption band in the IR spectrum at 2038 cm$^{-1}$. The resonances of the four diastereotopic protons of the two methylene units in the "wings" of the Cp$^C$ ligand are identified by their $^2J_{HH}$ coupling constants of about 13 Hz. For the two diastereotopic protons of the third methylene unit, a $^2J_{HH}$ coupling constant of 18 Hz is observed. One of these two protons shows an additional coupling ($^4J_{PH}$ = 6.8 Hz) to the phosphorus atom. The chemical shift of the $^{31}$P resonance of 51.7 ppm unambiguously proves the coordination of the phosphine site to the ruthenium(II) center. A peak at $m/z$ = 810 in the ESI mass spectrum corresponds to the mass of the fragment [M-CH$_3$CN]$^+$ a second peak at $m/z$ = 769 is assigned to the fragment [M-2 × CH$_3$CN]$^+$.

As mentioned above, the diphosphine functionalized Cp$^C$ derivative **3** can only act as a typical diphosphine donor. However, the two phosphine sites are linked by a chain of overall seven carbon atoms allowing either a chelating coordination of the diphosphine to one metal site or a bridging coordination to two metal sites. The simplest way of dealing with these alternatives is by coordination of two equivalents of gold(I) to compound **3**. The according gold(I) complex **7** was thus obtained in 83% yield in an analogous way as compound **4** using [(tht)AuCl] as the gold source (Scheme 6).

**Scheme 6.** Synthesis of the dinuclear gold(I) complex **7**.

In contrast to the mono nuclear gold(I) complex **4**, the dinuclear compound **7** is $C_2$ symmetric, which largely simplifies the $^1$H NMR spectrum, leading to only four resonances for the diastereotopic protons of the two chemically inequivalent types of methylene groups with coupling constants of about 12.5 resp. 17.5 Hz. The $^{31}$P NMR resonance at 24.9 ppm is close to the value obtained for compound **4** (24.4 ppm).

By the slow diffusion of *n*-pentane into a saturated toluene solution of compound **7**, single crystals suitable for an X-ray structure analysis could be obtained. Compound **7** crystallizes as colorless prisms of the triclinic space group P$\bar{1}$. Figure 4 shows the structure of **7** and summarizes relevant structural parameters.

Typical for gold(I) complexes, the two gold sites are coordinated almost linearly. To minimize steric interactions, in particular of the diphenylphosphinyl units, the two P-Au-Cl units point into opposite directions, which excludes an intramolecular Au⋯Au interaction. The bond parameters are in the same range as measured for complex **4**.

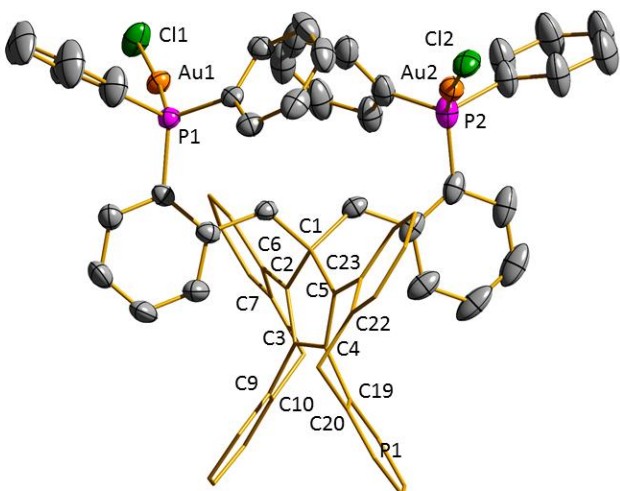

**Figure 4.** Molecular structure of compound **7** in the solid state. All hydrogen atoms are omitted for clarity. The central $Cp^C$ system is represented in stick-style for clarity, ellipsoids are drawn at the 50% level Characteristic bond lengths [Å], angles [°] and dihedral angles [°]: Au1-Cl1 2.2878(13), Au2-Cl2 2.2807(13), Au1-P1 2.2352(11), Au2-P2 2.2295(13), C1-C2 1.532(6), C2-C3 1.358(6), C3-C4 1.467(7), C4-C5 1.367(6), C1-C5 1.530(6), P1-Au1-Cl1 171.50(4), P2-Au2-Cl2 173.53(5), C3-C2-C6-C7 33.7(7), C4-C5-C23-C22 −34.8(6), C2-C3-C10-C9 49.9(10), C5-C4-C19-C20 47.5(6).

In 2006 Gelman et al. obtained a mono nuclear chelate complex as well as a dinuclear, chlorido-bridged palladium(II) complex by the treatment of the diphosphine ligand 1,8-bis(diisopropylphosphino)triptycen with different amounts of [PdCl$_2$(CH$_3$CN)$_2$] [52,53]. In a series of analogue reactions, ligand **3** was reacted with different amounts of [PdCl$_2$(CH$_3$CN)$_2$]. In contrast to the results of Gelman et al., only the dinuclear palladium complex **8** could be isolated, independent from the ligand-to-metal ratio and from solvent and temperature. By applying a slight excess of the metal source, the best yield was obtained (78%, Scheme 7).

**Scheme 7.** Synthesis of the dinuclear, chlorido-bridged palladium(II) complex **8**.

$^1$H and $^{13}$C NMR spectra of complex **8** clearly prove the presence of a $C_2$ symmetric compound. There are, e.g., only two doublets of doublets for the diastereotopic protons of the overall four methylene groups. While the methylene groups bridging the phenylene rings in the "wings" of the ligand give rise to two doublets at 3.82 und 3.89 ppm with a typical $^2J_{HH}$ coupling constant of about 12.4 Hz, the two methylene units linking the phosphine sites to the $Cp^C$ core are observed at 4.80 and at about 7.78 ppm. The value of the latter chemical shift is rather unexpected. It was identified by means of a H,H-COSY NMR experiment and is explained by the fact that of two of these methylene proton direct towards the central Pd$_2$Cl$_4$ unit. According to the $C_2$ symmetry of compound **8**, there is only one resonance in the $^{31}$P NMR spectrum, observed at 25.4 ppm, which is a typical value for this type of palladium(II) compounds [54].

By slow diffusion of *n*-pentane into a saturated toluene solution of complex **8**, single crystals suitable for an X-ray structure analysis could be obtained. Compound **8** crystallizes as colorless prisms of the monoclinic space group P2/$_n$ with two crystallographically independent molecules in the unit cell. Figure 5 shows the structure of one of these units of compound **8** and summarizes relevant structural parameters.

As expected on the basis of the NMR spectra of compound **8**, the phosphine donors coordinate to the central [(μ²-Cl)Pd(Cl)]₂ unit in an *anti*-conformation that preserves the $C_2$-symmetry of the Cp$^C$ moiety. The two palladium(II) sites occupy a distorted square-planar geometry. In comparison to structurally similar [(L)Pd(μ²-Cl)(Cl)]₂ (L = σ-donor ligand) compounds, the Pd···Pd distance (3.3098(6) Å) is by about 0.15–0.20 Å shorter, which we assign to the steric restrictions of the diphosphine ligand (see [55] and references cited therein). Due to the strongly σ-donating phosphine ligands the bond distances between the palladium(II) centers and the bridging chlorido ligands in *trans*-position to the phosphine donors are slightly longer than those in the *trans*-position to the apical chlorido ligands.

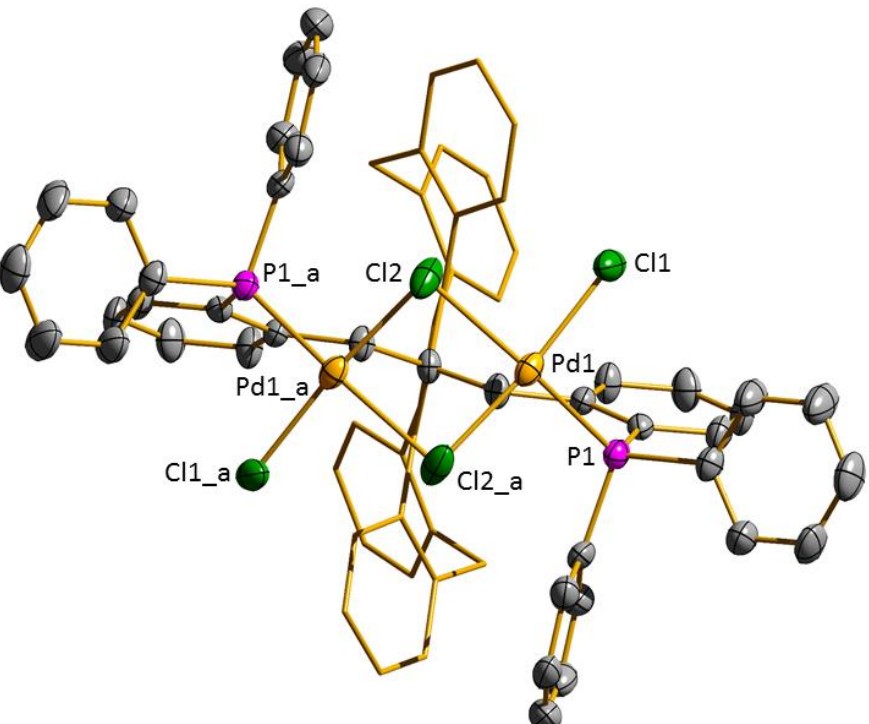

**Figure 5.** Molecular structure of compound **8** in the solid state. All hydrogen atoms are omitted for clarity. The central Cp$^C$ system is represented in stick-style for clarity, ellipsoids are drawn at the 50% level. Characteristic bond lengths [Å], angles [°] and dihedral angles [°]: Pd···Pd 3.3098(6), Pd1-Cl1 2.2880(16), Pd1-Cl2 2.3144(16), Pd1-P1 2.2499(14), C1-C2 1.533(6), C2-C3 1.360(7), C3-C4 1.475(7), Cl1-Pd1-Cl2 173.99(6), Cl1-Pd1-P1 89.96(5), Cl1-Pd1-Cl2_a 90.27(5), Cl2-Pd1-P1 95.71(5), Cl2-Pd1-Cl2_a 83.83(6), Cl2_a-Pd1-P1 171.23(5), C3-C2-C8-C9 35.5(5), C2-C3-C4-C13 47.1(6).

The fragment ClRh(CO) is well-known to prefer *trans*- over *cis*-coordination with chelating diphosphines as long as the linker between the phosphine sites allows this [56–63]. In ligand **3**, the two phosphine sites are linked by a chain of overall seven carbon atoms. Therefore, the formation of a *trans*-coordinated rhodium(I) complex seemed likely, when ligand **3** was reacted first with half an equivalent of [(μ²-Cl)Rh(CO)₂]₂ at room temperature. However, spectroscopic characterization of the product unambiguously proved the generation of the dinuclear rhodium(I) complex **9** (Scheme 8). Reaction with one equivalent of the precursor [(μ²-Cl)Rh(CO)₂]₂ provided complex **9** in excellent yields.

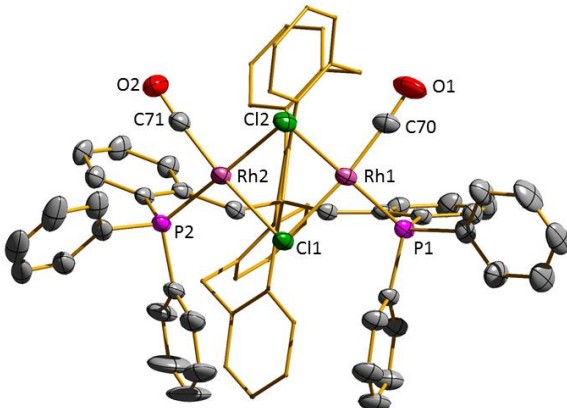

**Scheme 8.** Synthesis of the dinuclear, chlorido-bridged rhodium(I) carbonyl complex **9**.

The molecular structure of compound **9** resembles the structure of palladium(II) complex **8**. In contrast, in the case of **9** the phosphine donors are coordinated in a *syn*- and not in an *anti*-arrangement to the metal sites, which breaks the $C_2$-symmetry of the chelating ligand and leads to an overall $C_1$-symmetric structure.

This decrease in symmetry results in the typical pattern of resonances for the methylene protons as described above for other complexes with $C_1$-symmetry. As already observed for palladium complex **8**, the $^1$H NMR resonances of two of the methylene protons are shifted largely towards lower field (7.00 ppm). The two phosphorus atoms are magnetically not equivalent and show couplings to the neighboring $^{103}$Rh centers ($\delta$ = 35.9 ppm, $^1J_{RhP}$ = 171.5 Hz and $\delta$ = 39.8 ppm, $^1J_{RhP}$ = 175.0 Hz), which are close to values measured for other binuclear rhodium(I) complexes of similar structure [52,64,65]. In the $^{13}$C NMR spectrum two doublets of doublets at $\delta$ = 183.2 (dd, $^1J_{RhC}$ = 78.0, $^2J_{PC}$ = 20.6 Hz) and $\delta$ = 182.3 (dd, $^1J_{RhC}$ = 78.8, $^2J_{PC}$ = 21.1 Hz) are assigned to the rhodium-bound carbonyl ligands. In the IR spectrum of compound **9**, there are two carbonyl absorptions at 1991 and 1975 cm$^{-1}$ [52,64,65].

By the slow diffusion of *n*-pentane into a saturated toluene solution of complex **9**, single crystals suitable for an X-ray structure analysis could be obtained. Compound **9** crystallizes as yellow prisms of the triclinic space group P$\bar{1}$. Figure 6 shows the structure of **9** and summarizes relevant structural parameters.

**Figure 6.** Molecular structure of compound **9** in the solid state. The central Cp$^C$ system is represented in stick-style for clarity, ellipsoids are drawn at the 50% level. All hydrogen atoms are omitted for clarity. Characteristic bond lengths [Å], angles [°] and dihedral angles [°]: Rh⋯Rh 3.2797(5), Rh1-Cl1 2.3926(7), Rh1-Cl2 2.4127(7), Rh1-P1 2.2486(7), Rh1-C70 1.794(3), Rh2-Cl1 2.3880(7), Rh2-Cl2 2.4053(7), Rh2-P2 2.2431(7), Rh2-C71 1.805(3), Cl1-Rh1-Cl2 83.42(2), Cl1-Rh1-P1 96.09(2), Cl1-Rh1-C70 172.39(10), Cl2-Rh1-P1 176.67(3), Cl2-Rh1-C70 91.26(9), P1-Rh1-C70 89.53(9), Cl1-Rh2-Cl2 83.68(2), Cl1-Rh2-P2 97.62(3), Cl1-Rh2-C71 173.89(9), Cl2-Rh2-P2 177.68(3), Cl2-Rh2-C71 91.67(9), P2-Rh2-C71 86.90(9), Rh1-Cl1-Rh2 86.63(2), Rh1-Cl2-Rh2 85.80(2), C3-C2-C6-C7 38.2(3), C2-C3-C10-C9 46.6(4), C5-C4-C19-C20 45.7(4), C4-C5-C23-C22-37.7(4).

Both rhodium(I) sites in compound **9** occupy a distorted square-planar geometry, which is typical for d$^8$-configured transition metal ions and has already been observed for the palladium(II) complex **8**. In contrast to **8**, however, the phosphines are coordinating the two rhodium(I) centers in a *syn*-configuration. This is in agreement with the NMR data of the complex **9** that correspond to a $C_1$-symmetric structure. The Rh-Cl distances in *trans*-position to the π-accepting carbonyl ligands are slightly shorter than those in *trans*-position to the phosphine donors. The large interplanar angle of 132.28° between the chlorido-bridged rhodium(I) sites, the P-P distance of 6.378 Å, which is somewhat shorter with respect to the metal-free ligand and the relative long Rh-Rh distance of 3.28 Å confirm calculations carried out by López-Valbuena et al. on structural features of diphosphane ligands with large bite-angles [64].

As mentioned above, the fragment ClRh(CO) tends to form *trans*-coordinated mononuclear complexes with chelating diphosphines possessing long linker units. Therefore, ligand **3** was reacted with half an equivalent of the precursor [(μ$^2$-Cl)Rh(CO)$_2$]$_2$. In 2006, Gelman et al. reported the preferential formation of a monometalic chelate complex using shorter reaction times [52,53]. In our case, however, the selective formation of the monorhodium(I) complex **10** was achieved by combining a prolonged reaction time and an elevated reaction temperature (Scheme 9).

**Scheme 9.** Synthesis of the mononuclear rhodium(I) carbonyl complex **10**.

The $C_2$ symmetric ligand backbone in combination with a local $C_S$ symmetry of the coordination environment at the rhodium(I) site led to an overall $C_1$ symmetry and thus, as expected, to eight doublets for the methylene protons in the $^1$H NMR spectrum of compound **10**. Three of these resonances that are assigned according to their coupling constants to the methylene groups linking the Cp$^C$ backbone with the phosphine sites are strongly shifted towards lower field. The X-ray structure (see below) of compound **10** shows that these three protons are oriented towards the central (*trans*-P)$_2$Rh(CO)Cl unit. The $^{31}$P NMR spectrum of compound **10** reflects its $C_1$ symmetry: there are two doublets of doublets with a chemical shift of 26.2 und 35.6 ppm, each showing a large $^2J_{PP}$ coupling constant of 340 Hz and a smaller $^1J_{RhP}$ coupling constant (123.2 resp. 127.5 Hz). The large $^2J_{PP}$ coupling constant is typical for the *trans*-orientation of the two phosphine donors [66]. Compared to the chlorido-bridged complex **9**, the low field shift of the resonances of the two phosphorus atoms is not as pronounced. ESI mass spectrometry reveals a peak at $m/z$ = 1073.39, which corresponds to the cationic fragment [M-Cl]$^+$ (calcd.: $m/z$ = 1073.25). In the IR spectrum of compound **10**, there is an intense absorption $\tilde{v}$(CO) at 1961 cm$^{-1}$, which is typical for trans-coordinated, monomeric diphosphinerhodium carbonyl complexes [52,60–62,67,68].

By the slow diffusion of *n*-pentane into a saturated toluene solution of complex **10**, single crystals suitable for an X-ray structure analysis could be obtained. Compound **10**

crystallizes as yellow prisms of the triclinic space group P2$_1$/n; Figure 7 shows the structure of **10** and summarizes relevant structural parameters.

As reported for other mononuclear diphosphine carbonyl(chlorido)rhodium(I) complexes with long-chained diphosphine ligands, the two phosphine donors are oriented in *trans*-position [53,61–63]. The central rhodium(I) site is coordinated in a distorted square-planar geometry. Due to the steric influence of the Cp$^C$ moiety, the P1-Rh1-P2 angle is significantly smaller than 180° (167.27(3)°) as is the Cl1Rh1-C70 angle (169.08(9)°).

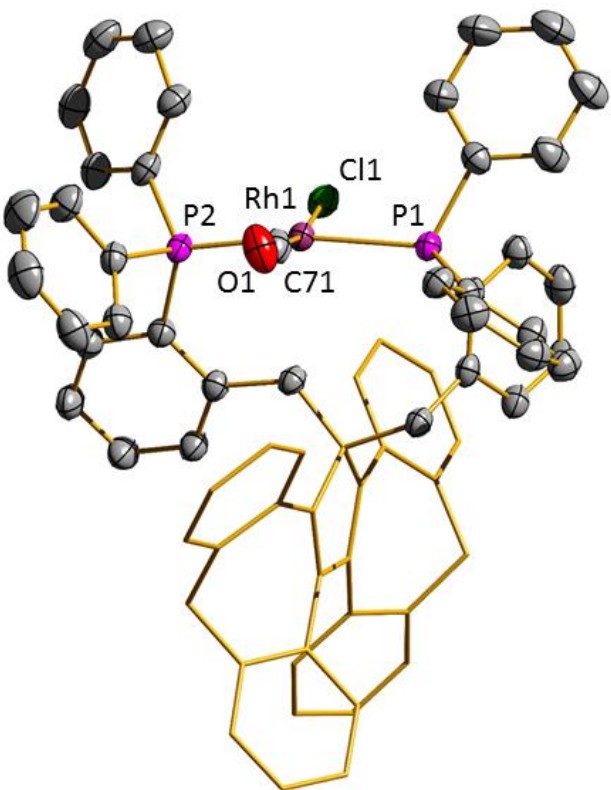

**Figure 7.** Molecular structure of compound **10** in the solid state. All hydrogen atoms are omitted for clarity. The central Cp$^C$ system is represented in stick-style for clarity, ellipsoids are drawn at the 50% level. All hydrogen atoms are omitted for clarity. Characteristic bond lengths [Å], angles [°] and dihedral angles [°]: Rh1-Cl1 2.3953(7), Rh1-P1 2.3081(7), Rh1-P2 2.3356(7), Rh1-C70 1.813(3), Cl1-Rh1-P1 87.04(3), Cl1-Rh1-P2 87.15(2), Cl1-Rh1-C70 169.08(9), P1-Rh1-P2 167.27(3), P1-Rh1-C70 90.38(8), P2-Rh1-C70 97.37(8), C3-C2-C6-C7 35.9(4), C2-C3-C10-C9 47.3(3), C5-C4-C19-C20 45.3(3), C4-C5-C23-C22 39.5(3).

## 4. Discussion

The novel chiral cyclopentadienyl-type ligand Cp$^C$H (**1**) is available following a high yield protocol of overall five steps starting from cheap dibenzosuberenone. We have shown in this manuscript, that Cp$^C$H (**1**) has turned out to be an ideal precursor for the synthesis of Cp-functionalized phosphine ligands. Both the monophosphine ligand **2** and the diphosphine ligand **3** are obtained by high yield nucleophilic substitution reactions with 2-[(chloromethyl)phenyl](diphenyl)phosphine using deprotonated compound **1**. In particular, the simplicity of the synthesis of **3** providing a long-chained and chiral diphosphine is of interest, since syntheses of similar systems usually suffer from poor yields.

With compounds **2** and **3** overall six different types of coordination could be realized. The monophosphine **2** still has a cyclopentadiene backbone with one proton attached to the five-membered ring and is thus able to coordinate like a typical phosphine or a typical cyclopentadienide or by combining these two modes. Examples of all three modes of coordination could be obtained by selecting the appropriate (transition) metal precursor. In contrast to monophosphine **2**, diphosphine **3** cannot be deprotonated anymore at the

cyclopentadiene site. It therefore reacts like a typical diphosphine. Bridging coordination to two gold(I) sites, as well as the *trans*-coordination to one rhodium(I) in a square-planar environment, are expected, in terms of the coordinative behavior of these two metal sites. In the latter case, a ten-membered ring is formed. However, we also found two examples for a coordination to two $d^8$ metal sites (palladium(II) and rhodium(I)) that are part of dinuclear chloride-bridged complexes. In these two cases, the ligand forms a twelve-membered ring including a $M(\mu^2\text{-Cl})_2M$ moiety. We assign this rather outstanding flexibility in coordination on one side to the long chain (seven carbon atoms) that connects the two phosphine centers. On the other side, there are four aromatic and one quaternary $sp^3$ hybridized carbon atoms in this chain, which reduce the number of possible conformations and thus stabilize the discussed modes of coordination.

In this manuscript, we have focused on aspects of the synthesis, spectroscopic data and structural elucidation of a series of complexes containing the new chiral phosphines **2** and **3**. The evaluation of, for example, the catalytical properties of selected compounds from this series is presently under investigation. Here, we concentrate on transformations that are known to require diphosphine ligands having a large bite angle and on transformations catalyzed by the *ansa*-bridged ruthenium complex **6**.

**Supplementary Materials:** The following supporting information can be downloaded at: https://www.mdpi.com/article/10.3390/chemistry5020062/s1. It contains figures of NMR and IR spectra of all compounds discussed in this manuscript.

**Author Contributions:** Investigation, F.N. and Y.S.; supervision and writing, W.R.T. All authors have read and agreed to the published version of the manuscript.

**Funding:** This research was funded by the Carl Zeiss Stiftung who donated a grant to F.N.

**Data Availability Statement:** Data is contained within the article and supplementary materials.

**Conflicts of Interest:** The authors declare no conflict of interest.

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
