# Peer review of "Phosphine Functionalized CpC Ligands and Their Metal Complexes"

_chemistry, doi:10.3390/chemistry5020062_

Round 1
Reviewer 1 Report
The paper by Thiel and co-workers reports on the nucleophilic substitution reactions of a rigid and sterically hindered cyclopentadiene ligands, and their reactions with various metal salts to display a wide range of coordination modes behaviour. The complexes obtained have been fully characterized using a number of techniques including MS, elemental analyses and single crystal X-ray crystallography. While there is no application reported for these metal complexes (given as a future work), I believe the diverse coordination chemistry illustrated here would be of interest to the synthetic and coordination chemists, and used as a platform for ligand modification. I thus recommend publication after minor corrections as stipulated below.
1. In the abstract, the authors should indicate the nature of substitution by giving the name of the compound (starting compound).
2. In the text, via, should be in italics.
3. In the syntheses of the Ru complex (6), it is interesting to note the displacement of the cymene ring from the Ru metal by the acetonitrile ligands/solvent molecule. One would expect the cymene to be more coordinating. Is this a kinetic product (excess NCMe) or a thermodynamic product? What would happen if the an innocent (non-coordinating) solvent like CH2Cl2 is used instead of acetonitrile? It is unfortunate that of all the complexes, only this one does not have an X-ray structure, possibly due to the lability of the NCMe molecule.
4. Is it possible to draw the ligands for example in Scheme 7-9 with the same configuration as in their respective complexes?
Reviewer 2 Report
In their paper, Närhig, Su and Thiel describe the a mized Cp/phosphine and a diphosphine ligand and a series of their complexes. The article is admirably written, shows an in-depth command of the literature which I expect will be welcomed by its readers. The quality and exhaustivity of the experimental section are equally praiseworthy.
The results presented in the paper are sound, backed by a careful analysis of the NMR data and supplemented for all complexes but one by XRD structures. I would suggest including (maybe in SI) a short overview of the crystallization methods that were attempted for compound 6, which might avoid duplication of efforts by another researcher.
The complexes are of significant interest for their potential catalytic properties, which are currently under investigation by the authors.
Overall, I recommend publication of the article as is, and apologize for the authors and editor for the delay in submitting my review.
Author Response
Please see the attachment. We have added our comment in blue color.

Reviewer 3 Report
The article reports the synthesis and characterization of a series of new phosphine functionalized cyclopentadiene ligands and their metal complexes. The amount of work is impressive, the evidence presented adequately justify the conclusions, and the article is definitely worthy of publication.
Minor points:
- The Scheme 1 caption should state which compound is given.
- L140. CH2Cl2 is redundant.
Author Response
The Scheme 1 caption should state which compound is given. -> we have done so. It's a general representation of the nuclei we were able to assign.
L140. CH2Cl2 is redundant. -> we have changed this.